# Tracking Down the Epigenetic Footprint of HCV-Induced Hepatocarcinogenesis

**DOI:** 10.3390/jcm10030551

**Published:** 2021-02-02

**Authors:** Tom Domovitz, Meital Gal-Tanamy

**Affiliations:** Molecular Virology Laboratory, Azrieli Faculty of Medicine, Bar-Ilan University, Safed 1311502, Israel; tomdo17@gmail.com

**Keywords:** hepatitis C virus (HCV), hepatocellular carcinoma (HCC), direct acting antivirals (DAAs), epigenetic signature

## Abstract

Hepatitis C virus (HCV) is a major cause of death and morbidity globally and is a leading cause of hepatocellular carcinoma (HCC). Incidence of HCV infections, as well as HCV-related liver diseases, are increasing. Although now, with new direct acting antivirals (DAAs) therapy available, HCV is a curable cancer-associated infectious agent, HCC prevalence is expected to continue to rise because HCC risk still persists after HCV cure. Understanding the factors that lead from HCV infection to HCC pre- and post-cure may open-up opportunities to novel strategies for HCC prevention. Herein, we provide an overview of the reported evidence for the induction of alterations in the transcriptome of host cells via epigenetic dysregulation by HCV infection and describe recent reports linking the residual risk for HCC post-cure with a persistent HCV-induced epigenetic signature. Specifically, we discuss the contribution of the epigenetic changes identified following HCV infection to HCC risk pre- and post-cure, the molecular pathways that are epigenetically altered, the downstream effects on expression of cancer-related genes, the identification of targets to prevent or revert this cancer-inducing epigenetic signature, and the potential contribution of these studies to early prognosis and prevention of HCC as an approach for reducing HCC-related mortality.

## 1. Introduction

Viruses rely on host machineries for transcription and translation and therefore disrupt cell functions to create a favorable environment for their propagation. In the case of oncogenic viruses, their proteins target specific cellular factors, and thus influence cellular signaling pathways which are often implicated in oncogenesis to promote viral persistence. However, the function of these oncogenes is usually not sufficient for cancer development, and other factors, such as chronic inflammation, are involved in and contribute to progression to cancer [1,2,3,4]. Indeed, Hepatitis C virus (HCV), which is the leading cause for hepatocellular carcinoma (HCC) in the western world, contributes to HCC development via both indirect host-mediated and direct HCV-mediated mechanisms [1,2,3,4]. HCV is a major public health problem, with over 70 million people infected worldwide and at risk for developing liver diseases [5]. According to the CDC, with the increasing number of new infections each year, the HCV-associated disease burden is expected to remain high in the next decade, even in developed countries, unless national programs aiming to reduce new HCV-infections will be accelerated (https://www.cdc.gov/hepatitis/policy/NationalProgressReport-HepC-ReduceInfections.htm).

HCV is an enveloped RNA virus which contains a single-stranded positive sense RNA genome of about 9.6 kb comprising a single open reading frame (ORF) that encodes a 3000 amino acid polyprotein precursor flanked by two untranslated regions (UTRs), a 5′ UTR and a short 3′ UTR. The amino-terminal, one-third of the polyprotein encodes the virion structural proteins: the core protein, and the envelope glycoproteins E1 and E2, followed by p7. The remainder of the viral genome encodes the nonstructural (NS) proteins NS2, NS3, NS4A, NS4B, NS5A and NS5B, which are located at the C-terminus and coordinate the intracellular processes of the virus life cycle [6].

HCV-associated hepatocarcinogenesis is multifactorial and involves the interplay of host, environmental, and viral factors. Complex virus-host interactions lead to multiple host responses, including the promotion of hepatic inflammation, fibrosis, and ultimately cirrhosis. Accordingly, HCV-associated hepatocarcinogenesis arises partially from insults to the hepatocyte accumulated during inflammation, including those caused by the generation of reactive oxygen species (ROS) [1,2,3,4]. Moreover, multiple HCV transgenic models develop HCC, supporting the direct oncogenic effects of HCV [2]. HCV proteins have also been shown to dysregulate processes, known as hallmarks of cancer, thus promoting cellular transformation [1,2,3,4]. Interestingly, although HCV is an RNA virus with a cytoplasmic life cycle, it still influences cellular processes, such as enforcing reprograming of host gene expression via epigenetic dysregulation. Compelling evidence was reported for dramatic changes in host chromatin structure imposed by HCV affecting the expression of tumor suppressor genes (TSG) or oncogenes, and therefore for the role of epigenetics in HCV-associated tumor initiation and progression [4,7,8,9,10,11]. The mechanisms involved in viral persistence leading to HCC, though, as well as the proportion of these mechanisms associated with virus-specific effects, are not fully understood.

Approval of direct acting antivirals (DAAs) has rapidly transformed the therapeutic landscape of HCV and provided remarkably high cure rates of over 95% sustained virological response (SVR) [12]. However, it is now evident that, as with the previous interferon (IFN)-based regimens, the risk of HCC persists after HCV-cure by DAAs [13,14,15,16,17,18,19,20]. Moreover, increased occurrence and recurrence of cancers following HCV treatment were also reported, although this phenomenon is not supported by other studies and remains debatable [21,22,23,24,25,26]. With national programs to improve the access to the new generation of antiviral therapies [27], exponential growth of the population of post-DAA-based SVR patients is expected. Therefore, post-SVR HCC is an acute emerging problem, and unveiling mechanisms leading to HCC post-cures, which are currently poorly defined, are of utmost importance. Such mechanisms may pave the way for clinical strategies for early detection of high risk for HCC progression after SVR, as well as post-cure intervention for HCC prevention.

Recent studies suggest that the long-term remaining of an HCV-induced epigenetic signature, even after cure of infection, is a potential mechanism for the persistence of HCC risk following SVR, and substantial evidence is provided to support this paradigm [7,8,9]. In light of this intriguing new concept, in this review, we provide a comprehensive overview of the evidence for HCV-induced dysregulation of cancer related genes and pathways via epigenetic interference. We specifically highlight the recent accumulated knowledge on the persistence of this epigenetic signature, discuss its implications for hepatocarcinogenesis and whether it is HCV-specific, and underline mechanisms of epigenetic imprinting by HCV infection that identifies potential targets for reversion of the epigenetic signature, which may be considered as a novel approach towards HCC prevention (Figure 1). Other recently suggested mechanisms that contribute to HCC progression following SVR by DAAs, such as the potential involvement of host immunity (Reviewed in [28]), are not discussed in this review.

## 2. Epigenetics and Gene Expression

Epigenetics is defined as potentially heritable elements in the genome that are not borne in the nucleotide sequence of DNA. These elements include post translation histone modifications (PTMs) and DNA methylation that collectively define the chromatin structure which is determined by the level of chromatin compaction. This structure may dynamically change from active to silent states and vice versa, resulting in activation or repression of gene expression [29]. At least eight types of modifications which modulate chromatin compaction are found on histones, such as acetylation, methylation and phosphorylation [30,31]. For example, methylation of the Lys9 and Lys27 residues on histone H3 (H3K9me2/3 and H3K27me3) is associated with gene repression, and acetylation on H3K4 (H3K4Ac) and H3K9 (H3K9Ac) is associated with gene activation [30,31]. Epigenetic alterations may result in the activation of oncogenes or inactivation of tumor suppressor genes, which contribute to malignant cancer hallmarks [29].

Each type of modification is executed by a group of specific enzymes. Histone acetylation states depend on the contradicting activities of histone deacetylase (HDAC) and histone acetyltransferase (HAT) enzymes. HATs are a family of at least 25 members, including HAT1, Gcn5/PCAF and p300/CBP, and HDACs include about 18 members that reverse the modification of acetylation [32,33]. Histone methylation is regulated by histone methyltransferase enzymes (HMTs) and DNA methyltransferases (DNMTs) alters the DNA methylation states. Hypermethylation of promotors results in gene repression and hypomethylation results in gene activation and expression [34]. Alterations in DNA methylation are involved, among other things, in gene expression, embryonic development, and genomic stability [35].

In addition to DNA methylation and PTMs, non-coding RNAs also dictate gene expression and are related as epigenetic regulators [36]. However, most of the epigenetic markers that were reportedly affected by HCV and remain persistent are DNA methylation and PTMs, and therefore these are the main focus of this review. Importantly, the epigenetic alternations discussed here include only those studied in HCV-infected samples in vivo and in vitro and do not include studies only performed on human HCC samples, in order to dissect the effect of the infection itself, from alterations that accumulate during carcinogenesis that may no longer represent viral-associated influence.

## 3. HCV-Induced Epigenetic Alterations and HCC

Viruses can navigate the chromatin structure through redirecting chromatin modifications and consequently altering host cell transcription that may promote oncogenesis [37,38]. In the case of HCV infection, both dysregulation of DNA methylation and histone modifications were described. Most of the studies focused on specific genes or pathway, and genome-wide profiles of these epigenetic alterations are scarce.

Alteration in host DNA methylation state was mostly reported in HCV core-expressing cells and was attributed to the well-known core-related increased expression levels of DNMT1 and DNMT3B. Studies demonstrated that HCV core induces regional hypermethylation of specific tumor suppressor genes, thus dysregulating key HCC-related pathways. One of the most studied cellular paths that was demonstrated to be epigenetically regulated by HCV is the cancer-related process of epithelial to mesenchymal transition (EMT). It was shown that EMT is induced by core expression through inactivation of WNT/β-catenin signaling. HCV-core protein epigenetically silences secreted frizzled-related protein (SFRP), extracellular WNT antagonist, by its promoter hypermethylation that enables the continued activation of the WNT signaling pathway. This methylation was suggested to be induced by DNMT1 (but not DNMT3A or DNMT3B), HDAC1 and methyl-CpG-binding domain (MBD) proteins that bind a region near the transcriptional starting site (TSS) of the SFRP1 gene. This contributes to HCC progression by increasing cell migration, which is abolished by SFRP1 re-expression or DNMT1 knockdown, in vivo and in vitro [39]. E-cadherin, which maintains cell-cell contact, also plays a crucial role in EMT and is epigenetically regulated by HCV-core, leading to its inactivation, and consequently, to cell migration and invasion. Studies found that E-cadherin promotor is hypermethylated by HCV-core protein in HCV core-transfected cells and in HCV infected cells [40,41]. Arora et al. suggested that E-cadherin promotor is hypermethylated by core through its upregulation of DNMT1 and DNMT3B levels [40]. Additionally, Ripoli et al. found that E-cadherin levels change due to an increased protein level of SIRT1, a class III histone deacetylase. Following treatment of HCV-core transfected cells with sirtinol, SIRT1 inhibitor, the hypermethylation of E-cadherin’s promotor was abolished and the levels of mRNA expression recovered [41].

HCV core protein also epigenetically affects hepatocyte apoptosis. H2O2 up-regulates P14 expression that induces MDM2 degradation in the proteasome, resulting in stabilization of p53. Core represses P14 expression by hypermethylating its promotor which inactivates P53 by MDM2. Therefore, H2O2 and core regulate the p14-MDM2 pathway, thus affecting P53 levels in an opposite way. This repression of P14 is completely abolished following treatment with 5-Aza-29dC, a DNMT inhibitor. P14 and P53 are tumor suppressors, and their inhibition contributes to HCC progression [42]. The repression of p14 by its promotor hypermethylation also antagonizes the function of trans retinoic acid (ATRA) which inhibits hepatocyte apoptosis. ATRA activates p14 expression via promoter hypomethylation, leading to the activation of the p53-dependent apoptotic pathway. Core and ATRA compete on p53 regulation by p14-MDM2 pathway, and in the presence of HCV core, the ability of ATRA to upregulate p53 and induce apoptosis was nearly abolished. The core-induced increase of DNMT1, DNMT3A, and DNMT3B levels was suggested to mediate the p14 promoter hypomethylation [43].

Core was also suggested to affect the cell cycle via methylation dysregulation. Core represses the expression of P16, a negative regulator of G1 checkpoint, by hypermethylating its promotor. This gene is inactivated in HCC pathogenesis leading to stimulation of cell growth and hepatocytes proliferation [44]. DNA methylation following HCV infection was also demonstrated to affect cell cycle via the growth arrest and DNA damage (Gadd45) gene family that encodes Gadd45β protein. This protein plays an important role in suppressing G2/M progression in response to genotoxic stress, and it was suggested to function in DNA excision repair. Higgs et al. found that Gadd45β expression is decreased in hepatoma cells bearing HCV replicons and in HCV infected cells. Interestingly, in liver biopsies from HCV-infected patients, a significant decrease in Gadd45β mRNA and protein levels was observed. Moreover, reduced Gadd45β expression was also detected in non-tumoral tissues from a transgenic mouse model expressing the entire HCV open reading frame. This reduction in Gadd45β expression is caused by HCV-induced promotor hypermethylation, and treatment with 5-azacytidine restored Gadd45β expression and cell cycle arrest. The aberrant cell cycle arrest may contribute to HCC progression [45].

Two distinct mechanisms for the core-induced upregulation of DNMT expression were proposed. Park et al. showed that core activates the AP-1 transcriptional activity by intracellular signal transducers such as ERK1/2, JNKs, and p38 MAP kinases and consequently, the promotor of DNMT1 that contains the binding motifs of the AP-1 complex. Core-expressing cells treated with a JNK inhibitor almost completely reverted the increased expression level of DNMT1 [44]. An alternative mechanism suggested by Arora et al. is related to core phosphorylation of Rb, which releases E2F1, thereby stimulating the Rb-E2F pathway and activating DMNT1 expression [40]. It is important to highlight that differences in mRNA and protein expression levels of DNMT1 and DNMT3B between HCV genotypes were found. For example, both are upregulated in genotype 1b HCV core expressing cells, and DNMT3B is upregulated at the protein level by genotypes 2a and 3a, but its mRNA levels did not change [46]. Importantly, it was demonstrated that DNMT1 and DNMT3B are required for HCV infection, protein expression, and RNA synthesis [47], suggesting that they play a role in the HCV life cycle. Interestingly, promotor hypermethylation and repression of E6-associated protein (E6AP) by core enables its evasion of poly-ubiquitination by E6AP and its proteasomal degradation in the cytosol [48].

In human samples, HCV-related hypermethylation was suggested to downregulate expression of the serine peptidase inhibitor/hepatocyte growth factor activator inhibitor type 2 (SPINT2/HAI-2) gene. More methylation in SPINT2/HAI-2 promotor was found in HCV-related cirrhosis that was increased in HCC patients, compared to the control group, pointing to progression in the methylation level during the process of carcinogenesis. Hypermethylation of SPINT2/HAI-2 is linked to carcinogenesis by inhibition of the HGF activator enzyme and thus the HGF/c-Met receptor pathway, as well as by downregulating the urokinase plasminogen activator, which may lead to extracellular matrix degradation, migration and metastasis. The dysregulated methylation was suggested to be related to inflammation or generation of ROS in HCV-infected cells, or due to direct alteration in epigenetic modifications by HCV [49].

So far, the information regarding the effect of HCV infection on global host DNA methylation is very limited. The genome-wide DNA methylation alterations following HCV infection were investigated using a mouse model with humanized livers. Interestingly, the authors did not find significant alterations in DNA methylation in HCV infected cells in cell culture, however, in HCV-infected mice, significant DNA methylation changes were observed. In mice, more genes were methylated following long-term infection (more than 16 weeks) than short-term infection (less than 16 weeks) suggesting that the methylation status dependents on duration of infection. Aberrant DNA methylation was found in numerous genes that are also methylated in HCC patients, such as RASSF1A, a well-known tumor suppressor gene, whose level of methylation increases as HCC progresses. The expression of IFNγ increased in HCV-infected mice in correlation with methylation status. IFNγ is expressed by natural killer (NK) cells and treatment with NK cell inhibitor resulted in reduced expression levels of IFNγ. Therefore, the authors suggested that DNA methylation is dependent on the induction of NK cell function in HCV-infected mice [50].

Evidence that global DNA methylation following HCV infection accumulates over time in the liver cells, leading to the methylation status found in HCC, were found in DNA methylation profiles of human liver samples. Wijetunga et al. compared DNA methylation alterations in HCV-infected liver biopsies, with control and HCC biopsies, to identify pre-neoplastic epigenetic and transcriptional events. In biopsies from HCV infected patients, the authors found an accumulation of global DNA methylation in enhancers that are active in liver cells and are enriched in binding sites of the transcription factors FOXA1, FOXA2 and HNF4A, which resulted in decreased expression of these genes. These enhancers are related to genes involved in liver cancer or stem cell development. The genes that are linked to stem cell development have increased CG dinucleotide density and polycomb-mediated repression, with elevate histone H3 lysine 27 trimethylation (H3K27me3) levels [10]. This study links pre-neoplastic transcriptional dysregulation of cancer related genes and DNA methylation with the generation of a pre-cancerous field effect that may lead to HCC development. Nishida et al. also investigated CpG loci methylation status in biopsies from HCV-infected patients and found that persistent viral infection accelerates methylation in the liver compared to normal liver. This cumulative methylation is similar to the methylation profile of HCC patients, suggesting that HCV infection could play a significant role in HCC progression [11].

In addition to DNA methylation, HCV infection also affects histone modifications, although this process is poorly understood. For example, HCV induces overexpression of protein phosphatase 2A (PPA2c), a serine/threonine phosphatase that plays a role in multiple cellular processes such as cell cycle regulation, cell morphology, development, signal transduction, translation, apoptosis, and stress response. Overexpression of PPA2c alters posttranslational histone modifications. For example, it inhibits methylation of histone H4 on arginine 3 and lysine 20, as well as acetylation on lysine 16, and thus it changes the expression of genes that contribute to tumorigenesis. Duong et al. found that PPA2c overexpression alters the expression of genes which are linked to hepatocellular transformation such as PTEN, IGFBP3, SOCS2, CDKN2A, FOXO1A, β-catenin, E-cadherin, c-myc, GADD45, and EGFR. In addition, PPA2c is up-regulated in liver biopsies from patients with chronic HCV, pointing to a link between HCV infection and HCC progression [51].

It was reported that HCV-core protein also regulates p300/CBP function and the acetylation state of histones or components of the transcriptional machinery. In core-expressing HEK and T lymphoma cells, its function in the nucleus increases HAT activity of p300 which induces histone acetylation and transcriptional activity, for example of NF-AT1, a transcription factor expressed in several cells of the immune system [52].

In HCV replicon cells, cytoplasmic and nuclear HDAC activity was higher than in control cells. Oxidative stress, induced by HCV infection, increases HDAC activity leading to alterations in chromatin structure by deacetylating transcriptional factors, binding sites and histones. In addition, this hypoacetylation inhibits binding of C/EBP and STAT3 to the hepcidin promoter, a negative regulator of iron absorption, resulting in decreased hepcidin expression. Hepcidin expression is repressed in HCC by promotor hypermethylation [53]. Treatment with antioxidants and an HDAC inhibitor restored hepcidin expression in replicon cells [54].

The effect of HCV infection on genome-wide alterations in status of histone modifications was only recently reported by us and others and will be discussed in detail in the next section.

In summary, these studies point for the role of HCV-induced alterations in DNA methylation and histone modifications in affecting cancer-related signaling pathways and in HCC development. However, the underlying mechanisms that govern the HCV-induced epigenetic alterations have yet to be fully deciphered.

## 4. HCV-Induced Epigenetic Signature Post-Cure of Infection

Epigenetic alterations, either due to normal developmental and differentiation processes or to environmental signals that affect the epigenome, are plastic on the one hand, but can also be maintained and imprinted in the genome and remain stable and heritable [55,56]. Therefore, we and others postulated that HCV-induced reprograming of the epigenome remains persistent after cure by DAAs, as a possible mechanism for HCC risk post-cure.

Evidence supporting this hypothesis was established by our and others’ recent publications that evaluated the genome-wide epigenetic alterations occurring following HCV infection and compared their profiles pre- and post-cure by DAAs [7,8,9]. As evident from these studies, HCV infection dramatically impacts the genome-wide positioning of histone modifications, thereby massively changing the gene expression patterns in the host cells. Strikingly, these changes were imprinted in the epigenome and persist even after cure by DDA treatment. This was observed in cell culture models, a human liver chimeric mouse model, and also in human liver biopsies.

In cell culture models (differentiated Huh7.5 cells infected with HCV), we demonstrated that HCV massively alters the epigenetic state of the host cell, through various acetylation and methylation modifications of histones. We observed significant changes in the profiles of three out of the four explored epigenetic markers: in the two active chromatin markers H3K9Ac and H3K4Me3 and the salient chromatin marker H3K9Me3. Thousands of altered regions in the genome occupied by these epigenetic markers were identified. The transcriptional changes positively correlated with the epigenetic changes. Importantly, of the genes that were altered, more than 50% remain persistently dysregulated after HCV cure by DAAs. Among 20 genes that were validated, 18 were not reverted following cure, both in RNA expression and level of histone modifications. Importantly, these changes were also observed in liver biopsy samples, pre- and post-SVR by DAAs [7].

Similarly, Hamdane et al. evaluated the epigenetic changes and their persistence following cure by DAAs in several HCV infection model systems: cell culture, a human liver chimeric mouse model, and human liver biopsies. The researchers performed genome-wide profile analysis of the histone modification H3K27ac, an epigenetic marker for active chromatin, and demonstrated significant repositioning of this epigenetic mark in HCV–infected compared to non-infected liver biopsies. Positive correlation was observed between the H3K27ac alterations in HCV-infected samples and DAA-treated samples, demonstrating the persistence of the epigenetic changes. Most of the alterations that are linked to TSGs and oncogenes were correlated with alterations in the expression of these genes. Similar results were obtained in the human liver chimeric mouse model that is permissive to HCV infection. The altered pathways were also persistent after cure by DAAs [8]. Jühling et al. investigated the epigenetic alterations and their translational values that are common to two risk factors for HCC: HCV infection and Non-Alcoholic Steatohepatitis (NASH). They also observed genome-wide repositioning of the histone marker H3K27ac in cell culture models for HCV infection (differentiated Huh7.5.1 cells) and for liver injury (co-culture of differentiated Huh7.5.1 and LX2 stellate cells treated with free fatty acids) and a positive correlation between the epigenetic status and transcriptomics and their persistence after cure [9].

The evaluation of the persistence of epigenetic alterations following IFN treatment have been inconclusive. In cell culture model, we showed that IFN-based cure more efficiently reverted both RNA and epigenetic markers levels to those comparable to uninfected cells [7]. This observation is in agreement with recent reports suggesting that HCC risk following DAAs treatment is higher compared to IFN-based treatment [57,58,59]. Other publications, however, do not demonstrate significant differences in risk of HCC development between the two treatments [20,60,61]. Supporting this data is the evidence from Hamdane et al., where the persistence of epigenetic changes following DAAs or IFN were positively correlated [8]. Thus, further research is needed to determine the effect of IFN-based therapy on epigenetic signature.

## 5. Implications of the HCV-Induced Epigenetic Signature for Hepatocarcinogenesis

An important question is whether this epigenetic signature induced by HCV plays a role in hepatocarcinogenesis. Previously, a 186 gene expression signature in 216 HCV-infected early stage cirrhotic patients that predict increased risk for HCC development, termed as prognostic liver signature (PLS), was identified [62]. Moreover, a 32 gene signature, which is a reduced version of the 186 gene signature, was evaluated in HCC patients with different etiologies (HCV, Hepatitis B virus (HBV), alcoholic liver diseases, Non-Alcoholic Fatty Liver Disease (NAFLD, now termed MAFLD- metabolic associated fatty liver disease), or NASH) and was demonstrated to be predictive of risk for HCC development in all etiologies [63].

These gene signatures were employed to assess the correlation between the epigenetic alterations and risk for HCC. Expression of this gene signature was reported to revert more in liver biopsies from SVR patients treated with DAAs that did not developed HCC, compared with those who developed HCC [63]. Hamdane et al. showed that the changes in H3K27ac epigenetic marker were related to genes associated with increased rates of liver disease and death within the previously published cohort of 216 HCV patients. The altered genes were also related to pathways intersected by the PLS genes. In addition, 900 of the altered genes are associated with the hallmarks of cancer [8]. We also analyzed data from the 216 cirrhotic patients with HCV infection and identified a signature of eight genes associated with increased risk for HCC development. We demonstrated that both the expression changes and the epigenetic changes observed in these genes in HCV-infected liver samples were persistent after cure by DAAs [7].

Pathway analysis of the epigenetic signature genes indicated that they are related to cancer. In cell culture, we showed that integration of the changes in H3K9Ac and gene expression alteration revealed that the altered genes are related to cytoskeleton remodeling (TGF, WNT), transport clathrin-coated vesicle cycle, cell cycle, development, immune response, B-Raf, NGF, mTOR/MAPK, lipid metabolism, endocytosis and membrane trafficking and epigenetic modifiers, among other cancer-related pathways [7]. Pathway analysis for the genes associated with the alteration in H3K27ac in HCV infected and cured liver samples also identified cancer and inflammation-related pathways, including TNFa signaling, inflammatory response, G2M checkpoint, epithelial–mesenchymal transition, phosphoinositide 3-kinase, Akt, and mammalian target of rapamycin [mTOR) [8]. Moreover, intersection of epigenetic modulation with concordant gene expression in HCV and NASH etiologies for HCC, identified genes that are changed in both etiologies, including oncogenes that were upregulated (FGFR1, CCND2, MLLT3, MAML2) and TSGs that were downregulated (FANCC, TSC2). The dysregulated genes in both etiologies were associated with HCC development and shorter survival in a cohort of HCV-infected patients. The PLS genes also showed epigenetic alterations that correlated in both etiologies. Among the genes that were changed in the two etiologies, 76% were also modulated in DAAs -cured patients that developed HCC [9].

Collectively, these analyses identified genes and pathways that are associated with HCC and therefore, suggest that the epigenetic alterations that are induced following infection with HCV and persist after cure are associated with hepatocarcinogenesis (Figure 1).

## 6. Is the Epigenetic Signature Specific for HCV Etiology of HCC?

HCV infection may lead to chronic inflammation and consequently fibrosis that may result in cirrhosis of the liver. In most cases, this process is critical for HCC development on the background of HCV infection, however, it is not obligatory. As discussed above, it is accepted that indirect, inflammation-derived as well as direct, virus-specific mechanisms contribute to HCC development in chronic HCV infections [1,2,3,4]. Since inflammation, fibrosis and cirrhosis may be caused by different HCC etiologies, such as HCV, HBV, alcoholic liver diseases, NAFLD (or MAFLD), NASH and aflatoxins [64], distinguishing between inflammation-derived or virus-specific HCC mechanisms may point for etiology-specific processes of carcinogenesis.

It is now evident that SVR following IFN-based or DAAs-based cure does not always lead to cirrhosis regression. Studies found a correlation between lower risk of developing HCC following DAA treatment in patients without fibrosis compared to patients with existing fibrosis and cirrhosis before treatment [16,65]. In addition, cirrhosis is correlated with higher mortality rates in DAAs-treated patients, which increases in correlation with the stage of fibrosis [65,66]. Therefore, an intriguing question is whether the epigenetic signature is caused by indirect or etiology-specific mechanisms.

In patient liver samples, Hamdane et al. demonstrated that a large fraction, but not all, of the epigenetic changes correlate with fibrosis stage. In addition, persistence of these changes was observed more in advanced fibrotic stages [8]. Jühling et al. also showed that the epigenetic dysregulation correlated with advance fibrosis. However, this association was observed in only 43% of the alterations. These studies suggest that the epigenetic changes are related only in part to fibrosis and cirrhosis and therefore point to the involvement of other forces contributing to the epigenetic dysregulation [9]. This conclusion is in line with reports on a small percentage of HCC development on the background of HCV infection without fibrosis.

Direct evidence for an HCV-specific mechanism of induction of epigenetic signature was established by identifying a fraction of this signature in inflammation-free models for HCV infection: cell culture, and an immune-deficient mouse model. We and Jühling et al. demonstrated that the epigenetic signature observed in cell culture models positively correlates with that observed in liver biopsy samples as well as with the PLS genes, and similar pathways were enriched [7,9]. These observations dissect the specific role of HCV in these changes and suggest that they are originated from specific HCV–hepatocyte interactions. These studies underscore cell culture models as a useful platform for studying the virus-specific epigenetic and gene expression signatures. Importantly, in an immune deficient human liver chimeric mouse model (uPA/SCID mice engrafted with human hepatocytes) that is permissive for HCV infection, with no detectable liver inflammation or fibrosis, alterations in H3K27ac were observed, as were strong correlations between these changes and transcriptional changes, similar to findings from patient samples. The authors identified altered genes that are common between the patient and mice samples and are persistently altered both epigenetically and transcriptionally. Therefore, these observations provide evidence that many of the changes are directly induced by the virus itself and not the inflammation it causes [8]. In addition, the fact that only human-specific reads were identified in the mice tissues in the applied ChIP-Seq pipeline validates that large proportion of the persistent alteration found in patients are originated from HCV-infected hepatocytes and not from surrounding non-parenchymal cells.

Another important observation is that a proportion of these epigenetic changes was HCV-specific, compared to other HCC etiologies. Hamdane et al. identified H3K27ac changes that are etiology-specific as well as changes that were common to HCV, HBV infection and NASH. Analysis of correlation of immune related genes between all etiologies resulted in low association, compared to all genes, suggesting that the similarities between etiologies are not due to inflammation alone [8]. Jühling et al. showed that the intersection of epigenetic modulation of H3K27ac with concordant gene expression in liver tissues with advanced fibrosis on the background of HCV and NASH identify genes that are changed in the two etiologies. The PLS genes also showed epigenetic alterations that correlated in both etiologies. Moreover, the gene expression pattern in a DEN/CDAHFD diet-induced mouse model of NASH-driven HCC was similar to the pattern observed in HCV and NASH patient samples [9].

Together, the reported data indicate that the epigenetic signature induced following HCV infection consists of alterations that are a result of inflammation and liver injury and alterations that are common to all HCC etiologies that are not inflammation derived, but also a significant proportion of this signature that is unique and specific for HCV etiology of HCC.

## 7. Possible Mechanisms of Epigenetic Imprinting by HCV Infection: Paving the Way for HCC Prevention?

The plasticity of the epigenome provides a unique opportunity to influence it and change unfavorable epigenetic states, for example with the use of epigenetic inhibitors for the treatment of various cancers [67]. In order to identify targets for chemo preventative therapy, it is important to understand the mechanisms that derive this epigenetic signature. Mechanisms of epigenetic memory that maintain an epigenetic state are not well understood. Generally, models suggest that during cell divisions histone modifications are reestablished by epigenetic “writers”, i.e., epigenetic enzymes (such as HATs), at positions that are marked by nascent histones containing a specific modification. It is known however that this process requires the continuous actions of the epigenetic enzymes in the S phase of the cell cycle (reviewed in [56,68]). Therefore, inhibiting the function of epigenetic enzymes may be a rational approach for achieving pharmacological reversion of dysregulated epigenetic states [69].

Recently, we and others evaluated the use of epigenetic drugs as means for reversion of the HCV-induced epigenetic signature post-cure. To evaluate the reversal of the epigenetic changes in H3K9Ac levels, we used histone acetyl transferase (HAT) p300/CBP inhibitor C646 that inhibits acetylation of H3K9. Treatment with this inhibitor reverted the transcriptional and epigenetic changes in the signature genes in HCV-cured cells by DAAs [7]. Similarly, Jühling et al. evaluated the potential of epigenetic inhibitors to revert the expression of PLS genes in HCV infected cell culture models using a panel of inhibitors targeting the action of various enzymes including HATs, bromodomain-containing proteins 3/4 (BRD3/4) which are readers of acetylated lysines, mixed-lineage leukemia protein/WD Repeat Domain 5 (MLL/WDR5) complexes required for the methylation of H3K4, and HDACs. Inhibitors for BRD3/4 and MLL/WDR5 reverted the PLS gene expression, and inhibitors for HAT and BRD3/4 reverted the HCV-induced EGF and NFkB2 overexpression, which are risk factors for HCC development. Moreover, knockouts of BRD3, BRD4 and HDAC9 prevented the HCV-induced overexpression of the PLS genes, demonstrating that their expression is associated with histone acetylation. Importantly, these inhibitors did not significantly affect HCV infection, supporting a direct effect of the inhibitors on epigenetic regulation, and not an indirect effect through altering HCV levels. These results indicate that since cell culture models demonstrate similar pattern of expression of PLS genes, they are a useful model for studying epigenetic and gene expression signatures and their reversion by inhibitors [9].

Similar results were observed in other models. In a DEN/CDAHFD diet-induced mouse model of NASH-driven HCC, the gene expression pattern was similar to the pattern observed in chronic HCV infection and NASH patient samples. Treatment with BRD4 inhibitor six weeks after NASH development reverted the expression of the PLS genes and pathways related to cancer. Importantly, the authors explored the potential of BRD4 inhibitor to revert HCC progression and found decreased liver tumor nodules, fibrosis and liver inflammation in this mouse model treated with the inhibitor. In addition, this inhibitor reduced the viability of HCC spheroids derived for human tissues [9].

The data reported so far strongly suggests that epigenetic drugs may be efficient in reverting the HCV-induced epigenetic signature and cancer-related processes. However, these inhibitors target enzymes that are essential for genome wide regulation of the epigenome and are not specific to HCV-induced alterations. To target changes in the epigenome that are HCV-specific, it is essential to uncover the upstream drivers of these changes that are induced by the virus and use inhibitors that are specific to the altered processes. However, the information regarding such paths is currently scarce.

We suggested the involvement of the EGFR signaling pathway as a mechanism that may induce epigenetic changes in HCV infected cells, for a number of reasons. First, previous reports demonstrated the activation of EGFR by HCV proteins [70,71,72,73,74,75]. We have recently demonstrated that HCV infection constitutively activates EGFR thus inducing invasion of HCV-infected cells [76]. Moreover, among the pathways that we have identified to be epigenetically altered, the top pathway was cytoskeleton remodeling, which is known to be induced by EGFR signaling [7]. Other evidence supporting this hypothesis showed the association of the EGFR pathway with cirrhosis, fibrosis, and HCC, and the reversion of the expression of cirrhosis signature genes by the EGFR inhibitor erlotinib [62,63,77,78,79]. Indeed, treatment with erlotinib reverted the expression and altered H3K9Ac levels for most of the signature genes in HCV-cured cells treated with DAAs. Moreover, treatment with the EGFR ligand EGF, that activates EGFR signaling, recapitulated the increased expression of most of the signature genes [7].

Another suggested HCV-induced mechanism that affects the epigenome is the unfolded protein response (UPR), which is known to be upregulated in HCV-infected cells. UPR inhibitor partially restored the dysregulation of the expression of HCC- high risk genes, while activation of UPR partially induced the expression of these genes, demonstrating that HCV-induced UPR induce epigenetic and transcriptional changes associated with HCC development [9].

In another study, transcriptome analysis of 523 fibrotic/cirrhotic liver tissues revealed dysregulated pathways that are associated with liver pathogenesis, which may induce the 32-gene HCC risk-predictive signature. Among these are the EGF, TGF-B, stress response, extracellular matrix, and interferon signaling. Importantly, the authors identified a novel druggable target for HCC prevention, the lysophosphatidic acid (LPA) pathway, that induces downstream pathways associated with HCC. They used specific inhibitors, ATX (AM063) and LPAR1 (AM095), to evaluate the chemoprevention of HCC development by targeting this pathway. These inhibitors prevented HCC development in a diethylnitrosamine (DEN) rat model for cirrhosis-driven HCC by suppressing LPA downstream pathways RhoA and ERK. In addition, in cultured human liver tissues, they also showed reversal of the gene signature following treatment with AM095 [63].

To conclude, inhibitors for epigenetic modifiers showed promising results as means for reversion of HCV-related epigenetic signature and HCC prevention. However, more research is needed to assess their effectiveness in clinical settings and to identify other inhibitors that target upstream virus-specific mechanisms affecting epigenome dysregulation (Figure 1).

## 8. Future Prospects

The incidence of HCC has been increasing dramatically worldwide over the past few decades, particularly in the West. In the United States, HCC has become the fastest-rising cause of cancer-related mortality ([80] and https://gco.iarc.fr/tomorrow). Modeling studies indicate that HCC incidence will continue to rise [17,81,82]. Now, with the new DAAs and the realization that the risk for HCC development persists after cure, it is clear that these projections are still valid, even in this new era of successful HCV therapy. The main curative treatment of HCC is surgery (resection or transplantation), but most patients are not candidates for this option. The multikinase inhibitor sorafenib is the standard of care for these patients [83]. New drugs are now approved for HCC, including the multikinase inhibitor Lenvatinib and the programmed cell death protein-1 (PD-1) immune checkpoint inhibitor Nivolumab. However, although these drugs have survival benefits, the response rate is low and short-term, usually few months (reviewed in [84]). Due to the current deficiency in treatment options for HCC, the overall survival of HCC patients remains poor [85]. Therefore, HCC prevention may be the most efficient measure to reduce HCC-related mortality, and therapies that inhibit liver disease progression and HCC development are urgently needed. The identification of mechanisms that maintain HCC risk following SVR provides the opportunity to develop diagnostic tools to predict cancer risk in SVR populations. Stratifying those who are at risk will enable identification of the groups that are most in need of interventions and treatments that may prevent progression to HCC, as a strategy to eliminate mortality associated with liver disease. The recently identified signatures of gene expression markers as well as epigenetic markers for post-SVR HCC risk discussed in this review open up a new area of research aiming to identify novel host targets for the development of drugs affecting the potentially reversible process of epigenetics, unlike the irreversible mutations leading to cancer. Targeting reversible cancer-related processes offers a new path for HCC prevention, even after HCV damage occurs, and provides opportunities for eliminating the persistent oncogenic activity of HCV after cure. Further research is needed to pin down specific upstream, druggable targets at the intersection of molecular pathways activated by HCV to alter the epigenome and transcriptome and develop HCV-etiology specific personalized interventions for HCC prevention.

## Figures and Tables

**Figure 1 jcm-10-00551-f001:**
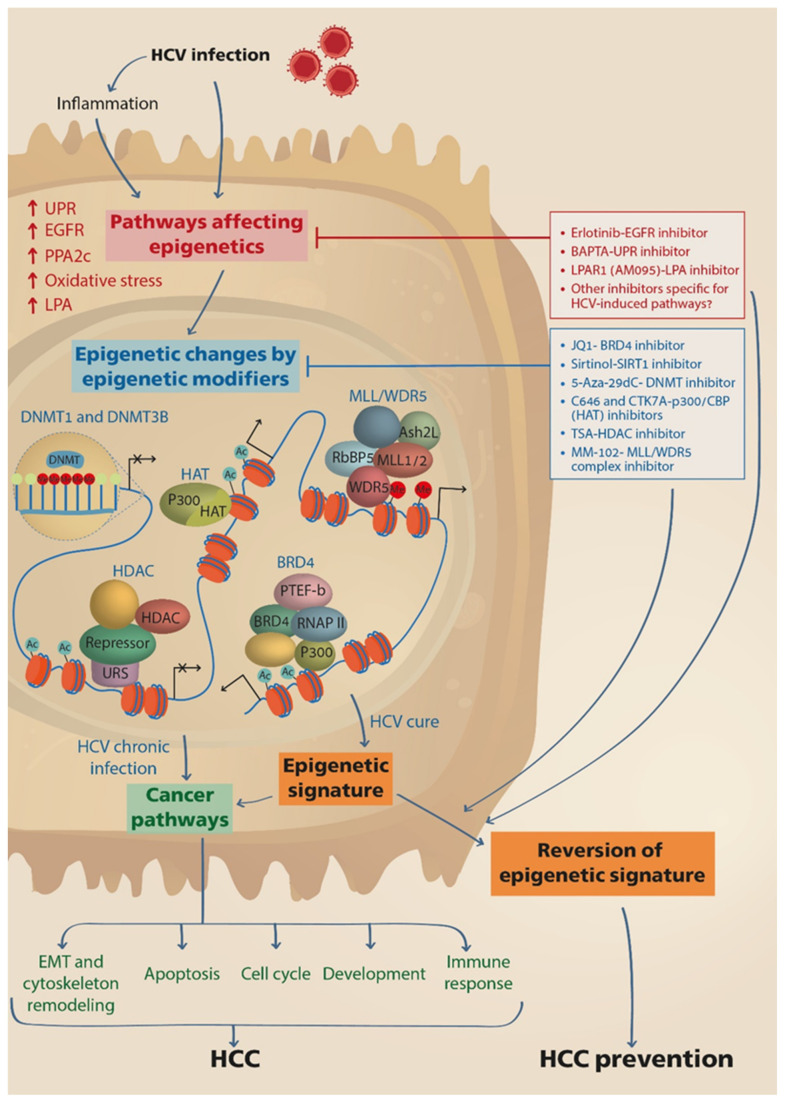
HCV induces epigenetic alterations that persist following cure with DAAs and may be reverted by drugs. HCV infection directly, or via induction of inflammation, stimulates cellular pathways that lead to the activation of epigenetic modifying enzymes and nuclear complexes, such as DNMTs, HATs, HDACs, BRD4 and MLL/WDR5 complexes, which execute genome-wide repositioning of PTMs and DNA methylation, and the reprograming of host gene expression. Consequently, pathways related to cancer are activated, contributing to HCC development and progression. The epigenetic changes remain persistent following cure of infection by DAAs therapy, thus maintaining the risk for HCC development. Drugs that prevent the induction of upstream pathways leading to activation of epigenetic modifiers or drugs that inhibit the function of the epigenetic modifiers downstream, may revert the epigenetic signature and prevent HCC development. DNMTs, DNA methyltransferases; HAT, histone acetyl transferase; HDAC, histone deacetylase; BRD4, Bromodomain-containing protein 4; PTEF-b, the positive transcription elongation factor; RNAP II, RNA polymerase II; URS, upstream repressive sequence; MLL, mixed-lineage leukemia protein; WDR5, WD Repeat Domain 5; RbBP5, RB Binding Protein 5; Ash2L, ASH2 Like.

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
