# Peer review of "Tracking Down the Epigenetic Footprint of HCV-Induced Hepatocarcinogenesis"

_jcm, 2021, doi:10.3390/jcm10030551_

Round 1

Reviewer 1 Report

Only minor comments to offer here. In general, this is a good review. One annoyance is that the epigenetic footprint data is somewhat scattered across different cell types in the liver, which muddles the focus in terms of specific cell gene expression profile. 

Some statements regarding the “induction” of cancer are questioned. Is it more prudent in some instances to state “association” with cancer development, rather than the former which suggests causation when such a link is not yet known.

Early in the intro, it is sometimes not clear if the author means to say that “in HCV infection” the expression of tumor suppressor genes occurs, or if they mean to say that distinct HCV factors are inducing these changes in gene expression. It is unclear if the cited references are meant to be showing that, unless we read them. The details subsequently unfold in section 3.

Authors occasionally refer to their own published works in this field, which further supports their credibility for authoring such a review, as they demonstrated their understanding of the state of the knowledge.

Overall, the paper is quite well written in its addressing of the technical aspects of how the virus may influence cancer development. For a review on the topic, it is sufficient the thorough.

Author Response

Only minor comments to offer here. In general, this is a good review.

  1. One annoyance is that the epigenetic footprint data is somewhat scattered across different cell types in the liver, which muddles the focus in terms of specific cell gene expression profile. 

Answer: In the revised manuscript we have added an explanation in the text that the data demonstrate that large proportion of the epigenetic changes are originated from specific HCV–hepatocyte interactions and from HCV-infected hepatocytes and not from surrounding non-parenchymal cells (page 9, 3rd paragraph).

  1. Some statements regarding the “induction” of cancer are questioned. Is it more prudent in some instances to state “association” with cancer development, rather than the former which suggests causation when such a link is not yet known.

Answer: In the revised manuscript the phrase “cancer-induced” was replaced with “cancer-associated” in the abstract and the introduction.

  1. Early in the intro, it is sometimes not clear if the author means to say that “in HCV infection” the expression of tumor suppressor genes occurs, or if they mean to say that distinct HCV factors are inducing these changes in gene expression. It is unclear if the cited references are meant to be showing that, unless we read them. The details subsequently unfold in section 3.

Answer: In the introduction, we summarize the data that will be detailed in the following text and therefore use “HCV infection” to include all that is known to occur following infection, although we do mention in the introduction that “HCV proteins have also been shown to dysregulate processes, known as hallmarks of cancer, thus promoting cellular transformation”. The text in section 3 we indeed provide detailed information on what is known about which of the HCV proteins trigger the changes in the cells, and the mechanism of action, although in most cases it is not known.

  1. Authors occasionally refer to their own published works in this field, which further supports their credibility for authoring such a review, as they demonstrated their understanding of the state of the knowledge.

Answer: So far, only our group and the group of prof. Thomas Baumert published comprehensive evaluation of the genome-wide epigenetic signature post cure of HCV by DAAs and its implications to HCC development, and therefore, the published work from the two groups are extensively discussed in this review.

Overall, the paper is quite well written in its addressing of the technical aspects of how the virus may influence cancer development. For a review on the topic, it is sufficient the thorough.

Reviewer 2 Report

The review submitted by Tom Domovitz and Meital Gal-Tanamy aims to summarize the evidence for HCV-induced epigenetic changes in the host genome which may impact the HCC risk after viral cure. The review is very well written and structured summarizing a topic, which is highly relevant to the field. Overall, the data is comprehensive and mostly up to date (see specific comments).

Specific comments:

(Minor)

  1. Abstract: The authors state that the incidence of HCV is increasing, however in the introduction no reference is given to this statement. The epidemiology introduced dates from 2015 – pre DAA. What are the estimates for the coming years?
  2. Future Prospects: some of the information seem outdated, i.e., the cancer statistics and the currently approved multikinase inhibitors in addition to sorafenib.
  3. The authors often cite other reviews. A citation of the original literature is preferential, e.g., page 2, line 17-20 “Compelling evidence was reported..”
  4. Page 2, last sentence “potential involvement of host immunity..”, (a) reference(s) should be provided.
  5. The term NAFLD should be replaced by the new terminology MAFLD (see PMID 32044314)
  6. Page 9, line 17 (paragraph): The authors discuss the inflammation-free animal models – is it the absence of fibrosis or the absence of inflammation in the livers of these animals? This should be briefly discussed.
  7. Activation of EGFR by HCV has been reported earlier. Additional references should be provided (e.g., PMIDs 18547392, 22855500).
  8. The header states Int. J. Environ. Res Public Health 2020 – this should be corrected to JCM
  9. Hyphenation page 3, line 37 “HCV core-expressing”
  10. Formatting errors when using Greek symbols throughout the manuscript (e.g., IFN-α, page 5, lines 29-30).
  11. Page 4, line 8: “Additionally” – not “Alternatively”
  12. Page 4, line 34: “genotypic stress” seems not being the appropriate expression (genetic stress?)
  13. Page 5, line 8: “downregulate” not down regulate
  14. Page 7, title 5.: “hepatocarcinogenesis” not “hepatocancinogenesis”
  15. Page 7, line 45: “Recently,” is not appropriate since the study was published 8 years ago
  16. Punctuation: Commas before and after .. ,i.e.,  .. or ..  , e.g., .. throughout the manuscript.
  17. Page 11, line 20: I assume that with “gene signature” the authors refer to the PLS – this should be specified here.

Author Response

The review submitted by Tom Domovitz and Meital Gal-Tanamy aims to summarize the evidence for HCV-induced epigenetic changes in the host genome which may impact the HCC risk after viral cure. The review is very well written and structured summarizing a topic, which is highly relevant to the field. Overall, the data is comprehensive and mostly up to date (see specific comments).

Specific comments:

(Minor)

1. Abstract: The authors state that the incidence of HCV is increasing, however in the introduction no reference is given to this statement. The epidemiology introduced dates from 2015 – pre DAA. What are the estimates for the coming years?

Answer: In the revised manuscript we have provided updated information from CDC regarding the increase of HCV incidence and the estimated for the future in the introduction: “According to the CDC, with the increasing number of new infections each year, the HCV-associated disease burden is expected to remain high in the next decade, even in developed countries, unless national programs aiming to reduce new HCV-infections will be accelerated (https://www.cdc.gov/hepatitis/policy/NationalProgressReport-HepC-ReduceInfections.htm).”

2. Future Prospects: some of the information seem outdated, i.e., the cancer statistics and the currently approved multikinase inhibitors in addition to sorafenib.

 Answer: In the revised manuscript we have added more updated references for the cancer statistics. Also, we have included information on new HCC drugs with updated reference (Future Prospects).

3. The authors often cite other reviews. A citation of the original literature is preferential, e.g., page 2, line 17-20 “Compelling evidence was reported..”

Answer: We have added citations of the original literature in page 2, line 17-20 as well as in other cases where it is possible. However, in cases where a text, that is not the main focus of this review, refers to high number of publications, we have maintained the citations of reviews that best summaries the referred topics.

4. Page 2, last sentence “potential involvement of host immunity..”, (a) reference(s) should be provided.

Answer: Since more than 20 papers report involvement of host immunity to HCC progression following SVR by DAAs, and since it is not the focus of this review, we have provided a citation for a recent review that summarizes what is known in the field (PMID: 33133084).

5. The term NAFLD should be replaced by the new terminology MAFLD (see PMID 32044314)

Answer: In page 8, first paragraph, we added the new terminology to the text: ” Moreover, a 32 gene signature, which is a reduced version of the 186 gene signature, was evaluated in HCC patients with different etiologies (HCV, Hepatitis B virus (HBV), alcoholic liver diseases, Non-Alcoholic Fatty Liver Disease (NAFLD, now termed MAFLD- metabolic associated fatty liver disease), or NASH) and was demonstrated to be predictive of risk for HCC development in all etiologies”.

6. Page 9, line 17 (paragraph): The authors discuss the inflammation-free animal models – is it the absence of fibrosis or the absence of inflammation in the livers of these animals? This should be briefly discussed.

Answer: In the revised version we added: “Importantly, in an immune deficient human liver chimeric mouse model (uPA/SCID mice engrafted with human hepatocytes) that is permissive for HCV infection, with no detectable liver inflammation or fibrosis, alterations in H3K27ac were observed, as were strong correlations between these changes and transcriptional changes, similar to findings from patient samples.”

7. Activation of EGFR by HCV has been reported earlier. Additional references should be provided (e.g., PMIDs 18547392, 22855500).

Answer: We have added to the text: “First, previous reports demonstrated the activation of EGFR by HCV proteins (PMIDs 18547392, 22855500,  25872741, 26886748, 19475692, 29705238).”

8. The header states Int. J. Environ. Res Public Health 2020 – this should be corrected to JCM

Answer: Corrected

9. Hyphenation page 3, line 37 “HCV core-expressing”

Answer: Corrected

10. Formatting errors when using Greek symbols throughout the manuscript (e.g., IFN-α, page 5, lines 29-30).

Answer: Corrected

11. Page 4, line 8: “Additionally” – not “Alternatively”

Answer: Corrected

12. Page 4, line 34: “genotypic stress” seems not being the appropriate expression (genetic stress?)

Answer: Corrected to “genotoxic stress”.

13. Page 5, line 8: “downregulate” not down regulate

Answer: Corrected

14. Page 7, title 5.: “hepatocarcinogenesis” not “hepatocancinogenesis”

Answer: Corrected

15. Page 7, line 45: “Recently,” is not appropriate since the study was published 8 years ago

Answer:  “Recently” was replaced with “previously”

16. Punctuation: Commas before and after .. ,i.e.,  .. or ..  , e.g., .. throughout the manuscript.

Answer: Corrected

17. Page 11, line 20: I assume that with “gene signature” the authors refer to the PLS – this should be specified here.

Answer:  “gene signature” was replace with “the 32-gene HCC risk-predictive signature”